# Effect of Prolonged Photoperiod on Light-Dependent Photosynthetic Reactions in *Cannabis*

**DOI:** 10.3390/ijms23179702

**Published:** 2022-08-26

**Authors:** Martina Šrajer Gajdošik, Antonia Vicić, Vlatka Gvozdić, Vlatko Galić, Lidija Begović, Selma Mlinarić

**Affiliations:** 1Department of Chemistry, Josip Juraj Strossmayer University of Osijek, Cara Hadrijana 8/A, 31000 Osijek, Croatia; 2Agricultural Institute Osijek, Južno predgrađe 17, 31000 Osijek, Croatia; 3Department of Biology, Josip Juraj Strossmayer University of Osijek, Cara Hadrijana 8/A, 31000 Osijek, Croatia

**Keywords:** industrial hemp, G-band, H-band, OJIP, TBARS, FT-IR

## Abstract

Industrial hemp is a fast-growing, short-day plant, characterized by high biomass yields and low demands for cultivation. To manipulate growth, hemp is usually cultivated under prolonged photoperiods or continuous light that could cause photooxidative damage and adjustments of photosynthetic reactions. To determine the extent of changes in photosynthetic response caused by prolonged light exposure, we employed chlorophyll *a* fluorescence measurements accompanied with level of lipid peroxidation (TBARS) and FT-IR spectroscopy on two *Cannabis* cultivars. Plants were grown under white (W) and purple (P) light at different photoperiods (16/8, 20/4, and 24/0). Our results showed diverse photosynthetic reactions induced by the different light type and by the duration of light exposure in two cultivars. The most beneficial condition was the 16/8 photoperiod, regardless of the light type since it brought the most efficient physiological response and the lowest TBARS contents suggesting the lowest level of thylakoid membrane damage. These findings indicate that different efficient adaptation strategies were employed based on the type of light and the duration of photoperiod. White light, at both photoperiods, caused higher dissipation of excess light causing reduced pressure on PSI. Efficient dissipation of excess energy and formation of cyclic electron transport around PSI suggests that P20/4 initiated an efficient repair system. The P24/0 maintained functional electron transport between two photosystems suggesting a positive effect on the photosynthetic reaction despite the damage to thylakoid membranes.

## 1. Introduction

Industrial hemp (*Cannabis sativa* L. subsp. *sativa*) is one of the oldest cultivated plants used for many purposes throughout centuries. It originated from Eurasia and was spread worldwide gaining its popularity as a source of fibers and seeds. Due to the introduction of other fiber sources, as well as concerns regarding its psychoactive properties, hemp cultivation progressively declined in the 20th century [1]. The constantly increasing demand for renewable raw materials, especially in the bio-feedstock production and biorefineries, and changes in the legalization of growth, resulted in the comeback of industrial hemp production in recent decades [1,2]. In 2020, the Global Industrial Hemp Market size was over USD 205 million with estimated compound annual growth rate (CAGR) over 6% in the period between 2021 and 2027 [3]. Industrial hemp varieties contain low levels of psychoactive substance delta-9-tetra-hydrocannabinol (THC) [1]. According to the current European Union legislative, hemp cultivars allowed for production may not contain more than 0.3% THC on a dry basis [1,4]. However, higher content of non-psychoactive cannabinoids such as cannabidinol (CBD), cannabidiol (CBG), and cannabichromene (CDC), is a reason for the growing interest in hemp production for the use in cosmetics and pharmaceuticals [5,6].

Light is one of the crucial environmental factors influencing growth, development, and biosynthesis of phytochemicals in plants [7]. The Earth’s rotation on its axis results in regular light–dark cycles. The length of the light period is termed as the photoperiod. As sessile organisms, plants have developed sensitive mechanisms to measure and to adapt to the length of the photoperiod [8,9]. The duration of the photoperiod, in combination with temperature, light quality, and intensity, regulates plant growth and development through various internal and external signals [10]. The detection of light in plants is mediated by photoreceptors. These are divided into families according to the spectrum of the solar light they sense; red and far-red lights are absorbed by phytochromes, blue light and UV-A by cytochromes, phototropins and F-box containing flavin-binding proteins, while UV-B light is sensed by the UVR8 photoreceptor as recently shown [9,11]. Additionally, by modifying the composition of wavelengths, activity of phytohormones may also change which can stimulate flowering, inhibit elongation of the stem, or reduce plant height [10]. Chloroplasts also have a role in plants’ light sensing by altering their ultrastructure in response to different photoperiods [9].

Considering the effect of photoperiod on flowering, plants can be divided into: long-day plants, which flower under a longer photoperiod; short-day plants, which flower under a shorter photoperiod; and day-neutral plants, which are not responsive to changes in photoperiod [8,9]. Besides flowering, photoperiod also influences daily photosynthesis, growth, and starch metabolism [9,12]. The quality and quantity of light affects the excitation of PSI and PSII, key components in photosynthetic processes [13]. It was shown that photosynthetic capacity is increased by longer photoperiods which in turn induces growth [14]. However, numerous negative and adverse effects can be also induced in conditions of continuous light or prolonged photoperiods, such as chlorosis, decrease in plant growth and its productivity and yield, high starch production, and stress induction [10]. Plants grown under long days contain chloroplasts with smaller grana stacks and increased chlorophyll content. Photoperiod can also influence plants’ resistance to drought and salt stress and, according to recent evidence, the length of light period can have an impact on the response to pathogen infection [9,15].

The effects of the continuous light on the development and growth of plants have been explored but often revealed contradictory responses in plants [16,17]. Some plants have shown increased productivity when grown under continuous light, which can positively influence the greenhouse cultivation [7,16]. However, when exposed to continuous light, some plant species developed leaf chlorosis followed by reduced plant growth and yield. The reduced rate of photosynthesis observed under continuous light was related to the increase in leaf starch that was a consequence of the reduced carbohydrate export. This over-accumulation of carbohydrates can lead to a reduced amount of electron acceptors followed by photosynthetic electron transport chain donating electrons to O_2_. During this process, reactive oxygen species (ROS) are generated, consequently causing oxidative damage [10,16,17]. Photooxidative stress, created by higher production of ROS, induces elevated activity of antioxidative mechanisms including soluble and lipophilic low-molecular-weight antioxidants, detoxification enzymes, and different repair mechanisms [17,18]. Singlet oxygen (^1^O_2_) is believed to be the major ROS produced in plant cells under continuous light. Lipid-soluble compounds present in thylakoid membranes, carotenoids, and various prenyl lipids, participate in the protection against ^1^O_2_ terminating lipid peroxidation chain reactions [18,19].

Hemp is a fast-growing, short-day, annual plant, characterized by high biomass yields and low demands for cultivation [2,20]. It can adapt to different climates and the growth is influenced by the temperature, solar radiation, nutrients, and water availability [1,2]. The light quality and quantity play an important role when cultivating cannabis in a controlled environment. Hemp cultivars have exhibited diverse photoperiod requirements for vegetative growth and reproductive development [21,22]. Its capacity to convert photosynthetically active radiation into biomass is exceptionally efficient [23]. Longer photoperiod favors photosynthesis, thereby increasing plant growth and dry weight, but it delays flowering. Under a continuous light regime of 18 or 24 h, hemp plants remain in the vegetative state. Reproductive growth is initiated after a two-week exposure to 12 h of continuous light daily [6,22,24]. However, it has been indicated that some varieties can flower even under a prolonged period of light [6]. Different wavelengths of light can also affect plant morphology, physiological processes, and development. The most common light source used nowadays for indoor cannabis production is light emitting diodes (LEDs) which can have a widely varying spectrum [23,25]. Recent findings suggest that blue light increased fresh and dry biomass in industrial hemp, plant height, and number of leaves per plant [25] while the combination of red and blue light resulted in smaller leaf area and more compact morphology in comparison to a white light source [26,27].

Although the effect of light quality and duration on indoor grown plants is the topic of numerous studies, mechanisms of adaptations to photosynthetic reactions to overcome excess energy feed are scarce. Hemp growers regularly cultivate *Cannabis* under prolonged photoperiods or even under continuous light to manipulate growth and promote flowering at the expense of running normal physiological processes. In addition, to induce optimal growth outcomes, indoor cultivation usually involves changes in light sources and wavelength composition. Therefore, we hypothesized that in such conditions plants experience stress and adjustments of photosynthetic reactions are necessary to avoid deleterious effects of accelerated plant growth. Prolonged exposure to light would induce functional and structural changes of components in the photosynthetic apparatus in order to run competent photosynthetic reactions. Thus, we aimed to determine the effect of different durations of light exposure as well as different types of light on primary photosynthetic reaction complemented with the level of lipid peroxidation as a measure of oxidative stress in two *Cannabis* cultivars.

## 2. Results and Discussion

### 2.1. Fast Chlorophyll a Fluorescence Transients

Fast chlorophyll *a* fluorescence (ChlF) measurements were performed on both *Cannabis* cultivars grown under purple (P) and white light (W), exposed to 20/4 and 24/0 photoperiods and compared to the 16/8 photoperiod as a control. The OP normalized curves (Figure 1A,F) showed major differences between treatments at J and I steps for both cultivars. The ΔVt was calculated as the difference between photoperiods under white and purple light for each cultivar individually, and it was used as a reference for data normalization. The ΔV_OP_ (Figure 1A,F) revealed additional differences in range from 0.02 to 30 ms, which corresponds to L-, K-, and H-bands with amplitudes more pronounced in the USO31 cultivar. Positive amplitudes are usually associated with the nutrient deficiency [28,29], heavy metal stress [30], salt stress [31,32], drought [33], and low temperature [34]. Growing plants at W24/0 caused the appearance of positive amplitudes in both cultivars, with the exception of a negative peak around 30–300 ms, which corresponds with the G-band. Reduction at the 30–300 ms peak reflects the ability of re-reduction of plastocyanin and P700* in PSI [35], meaning that prolonged exposure to light led to the loss of capacity for photochemical reduction of Q_A_ and, consequently, lowered the transfer of electrons to the PSI acceptor side [36].

The differential curves for individual bands, L-, K-, H-, and G-band, enabled detailed analysis regarding their position and the relation to the main steps as well as to the events of primary photochemistry [37]. Significant changes were induced by growth at different photoperiods under white and purple light in both cultivars. The L-band, derived from ΔV_OK_ kinetics, enables a comparative study of energetic connectivity between the samples. In Finola, the L-band (Figure 1B) was positive only for plants grown at W20/4. However, in USO31 all treatments induced positive L-bands (Figure 1G) with higher amplitudes observed for 20/4 photoperiods, regardless of light source. Positive L-band is an indicator of low energetic connectivity of PSII units [29,38] suggesting that the 20/4 photoperiod lowers light absorption and diminishes its utilization in the initial step of photosynthetic process. Additionally, it can be observed that 20/4 photoperiods in USO31 shifted the band peak to the right. A longer time needed to reach the maximum peak at the 20/4 exposure compared to the 24/0 photoperiod suggests an even slower energy transfer from antennae to the reaction centers (RC). Negative L-bands recorded in Finola for 24/0 photoperiods, regardless of the light source, as well as for P20/4, suggested efficient utilization of absorbed light energy into the photochemical reaction and also the greater stability of the system [38]. Better grouping and connectivity between PSII RCs was reported recently in Japanese knotweed plants grown under fluctuating light [39] as well as in drought-treated barley [40] and perennial grasses grown in heavy-metal-contaminated soil [41]. It was suggested that plants that can resist the decrease in connectivity between PSII antennae in stressful conditions could provoke the ability to resist changes in stacking/structure of the thylakoid membranes. Therefore, prolonged exposure to light conditions (20/4 and 24/0) in Finola cultivar contributes to the maintenance of the thylakoid membrane integrity.

The K-band plotted as the difference kinetic ΔV_OJ_, is a measure of the oxygen evolving complex (OEC) functionality and/or functional PSII antennae size [38] and it is suggested to be a reliable indicator of stress in plants even before visible symptoms occur [40]. In the Finola cultivar, W20/4 and P20/4, as well as P24/0 induced positive K-band (Figure 1C). However, more pronounced amplitude was observed for P20/4 compared to other positive bands. The USO31 cultivar (Figure 1H) also showed positive amplitudes of K-band when grown at 20/4 under both light types. A positive K-band can be induced by various types of stress, such as salt stress [31], drought [36,40], increased temperature [42,43], and nutrient deficiency [29]. When positive, the K-band reflects an inactivation of OEC and/or increase in size of functional PSII antennae [38]. More pronounced bands, as observed in Finola P20/4 as well as in USO31 P20/4 and W20/4, can be associated with uncoupling of OEC that leads to the imbalance between donations of electrons from OEC to the oxidized chlorophyll at the PSII RC [29,31]. Such results indicated that prolonged exposure to light caused disturbance to the electron transport from OEC toward the PSII suggesting photoinhibition and inactivation of OEC. This in accordance with the result of %OEC (Figure 2A,B) which was shown to be decreased under the same treatments (W20/4 and P20/4) for both cultivars. However, the 24/0 photoperiod generated negative amplitudes in the USO31 cultivar grown under both light types. The negative K-band is usually sign of tolerance to various stressful conditions [32,44,45,46,47]. The accessibility of several alternative electron donors to OEC, such as ascorbate [48] or proline [49], can act as an alternative source to replace a sufficient number of electrons to be sent in the direction of PSII in order to drive efficient photosynthetic reactions. It was also suggested that exposure to continuous light could lead to OEC interruption, but the use of alternative electron donors could be an efficient photoprotective strategy [50]. Therefore, it could be assumed that OEC in USO31 at W24/0 and P24/0 ensured availability of alternative electron donors to achieve adaptation to prolonged light exposure.

The difference kinetics normalized between J and I steps (ΔV_JI_) can be distinguished as the H-band, while IP normalization (ΔV_IP_) reveals the G-band. The J-I-P part of the transient is considered as thermal phase of the induction curve and it describes multiple turnover events [37]. The shape and amplitude of the H-band is related to the reduction and oxidation of the plastoquinone (PQ) pool [36,51]. It was suggested that elimination of restrictions on the PSI acceptor side or stimulation of cyclic electron transport around PSI would lead to reoxidation of the reduced PQ pool [52]. In our investigation, the Finola cultivar showed a positive H-band (Figure 1D) at P20/4 and P24/0. A positive H-band was also observed in USO31 at W20/4 (Figure 1I). A constant flow of electrons from PSII results in a higher reduction of the PQ pool and in a decrease in its capacity [53]. Accumulation of reduced electron carriers such as PQ, Cyt_f_, and plastocyanin causes formation of a positive H-band [45]. Finola plants grown at W20/4 and W24/0, however, revealed negative inflexion at the initial part of the curve and a positive peak at the end of the curve. Such fluctuations showed changes in the PQ pool size that goes from bigger to smaller, suggesting that white light caused a fast reduction in PQ. Similar inflected curves were reported for *Cuscuta*-infected *Ipomoea tricolor* plants [54]. On the other hand, negative H-bands indicate slowing down of electrons from water to PQ molecules [45] and, consequently, slower reduction of the PQ pool [53]. Our *Cannabis* cultivar USO31 exhibited reverse inflexions compared to Finola; the W24/0 and P24/0 showed positive initial parts followed by small negative end inflexion, with P24/0 plants shifted to more negative amplitude (Figure 1I) suggesting changes in PQ pool size from smaller to bigger due to slowing down of the reduction rate.

The shape and the size of the G-band unveils the reduction rate of the PSI acceptor side by the electron driven out of the PQ pool and depends on the number of available NADP^+^ [29]. It provides information on the relative size of the PSI acceptors pool while the shape of the band could be structured by the efficiency of electron flow and the PSI reduction rate [51]. In our study, a positive G-band was observed in both cultivars in P20/4 plants with higher amplitude being measured in the Finola cultivar (Figure 1E,J). Such results suggested that the size of the PSI acceptor pool, as well as its functionality, decreased and that this electron transport at the PSI acceptor side impediment was greater in Finola plants. Similar results were reported for high-density-planted maize [55] and for *Ipomoea* plants infected with *Cuscuta* [51]. In contrast, a negative G-band was detected at W24/0 in Finola and at P24/0 and W24/0 in USO31. A negative band is a sign of a larger end acceptor pool that requires slower rising to the maximum. Zagorchev et al. [51] suggested that a negative G-band points to the increased activity of PSI, while Kalaji et al. [29] implied that in suboptimal conditions plants develop a compensatory mechanism by increasing the number of available NADP+ molecules per active RC. In our case, prolonged exposure to light (24/0 photoperiod) as an unfavorable growth condition, might induce development of such compensatory mechanism.

### 2.2. Analysis of JIP-Test Parameters

Selected JIP-test parameters, presented as radar plots (Figure 2), give us insight into effect of different photoperiods and growth light on photosynthetic reactions in two *Cannabis* cultivars, Finola (Figure 2A) and USO31 (Figure 2B). All parameters are normalized to their respective controls measured at the 16/8 photoperiod for each growth light and cultivar individually. The performance index for energy conservation to the reduction of intersystem electron acceptors (PI_abs_) is well known to be a very sensitive parameter demonstrating the vitality of photosynthetic units [38]. Our results showed that all plants experienced significantly decreased PI_abs_, with the exception of Finola P20/4 that showed no differences compared to the control. Such results suggested that alteration in duration of light exposure downregulated PSII function. This PI_abs_ decrease was mostly accompanied with a decrease in maximum quantum yield of primary efficiency (φ_P0_), additionally confirming the loss of photosynthetic efficiency. It was suggested that continuous light could increase vitality and efficiency for using excess light for electron transport beyond PSII and for carbon assimilation [50]. However, this was not the case for *Cannabis* cultivars. It was found that decrease in φ_P0_ exposed to salt stress was the result of inhibited redox reactions after Q_A_ which caused additional impairment of electron transport between Q_A_ and Q_B_ [56].

Compared to the control, white light in both cultivars and photoperiods triggered significant decreases in efficiencies for electron transport further than Q_A_^−^ (ψ_E0_) and efficiency to reduce the end acceptor at the PSI acceptor side (δ_R0_) as well as corresponding quantum yields, φ_E0_ and φ_R0_. Such results indicate a lower ability for regulation of absorbed energy and its utilization and dissipation by photosynthetic membranes [31]. On the other hand, purple light did not induce such changes. In Finola, P20/4 increased ψ_E0_, while δ_R0_ and φ_R0_ decreased, suggesting that such conditions caused diminutive leakage of electrons at the PSI acceptor side, most probably due to the inactivation of ferredoxin NADP^+^-reductase [56]. This is additionally corroborated with the previously mentioned appearance of a positive G-band (Figure 1F). Further, in Finola P24/0, φ_E0_ decreased and δ_R0_ increased, while the USO31 cultivar showed an increase for δ_R0_ and φ_R0_ only under the P24/0 treatment while other efficiencies and quantum yields remained at the same level as in the control plants. Increase in δ_R0_ and φ_R0_ can be explained as an increase in electron transport extent between two photosystems which could be the result of a decreased ratio of active RCs in PSII and PSI [28]. Therefore, such results indicated highly efficient PSI electron transport suggesting light-dependent changes in PSI activity. It was generally proposed that enhancement of PSI activity could be the physiological adaptation to stressful conditions that would lead to the increase in PSI reduction rate [57]. Such improvement in PSI activity might be also the result of formation of cyclic electron flow around PSI [58] that could alleviate the photodamage of PSI by regulating the PSII activity. Therefore, in the condition of prolonged exposure (24/0) to purple light in the USO31 cultivar, the distribution of excitation energy between PSII and PSI was highly regulated and, therefore, served as protective mechanism.

Changes in the flux ratios and specific fluxes per active RCs reveal accurate estimation of primary photosynthetic processes after growth under prolonged light conditions. White light provoked specific changes that were alike for both cultivars, however, there was a slight difference related to the duration of light exposure. Thus, W20/4 induced an increase in absorption (ABS), trapping (TR_0_), and dissipation (DI_0_) per active RC. While the electron transport further than Q_A_^−^ (ET_0_) remained at the same level, it was followed by the decrease in electron flux that reduced final electron acceptors at PSI (RE_0_) compared to control. On the contrary, exposure to an even longer duration of light (W24/0) induced only an increase in DI_0_/RC with a subsequent decrease in RE_0_/RC in both cultivars compared to the control. Increases in ABS/RC and TR_0_/RC while φ_P0_ decreases indicate that certain parts of RCs were inactivated as a result of incapacitated OEC (Figure 1C,H). There is also a possibility that a certain part of active RCs is transformed into silent ones that are capable of efficiently absorbing excitation energy, but are unable to reduce Q_A_ and instead excess excitation energy is mostly dissipated as heat [38,59]. This is corroborated with the fact that DI_0_/RC also increased in both *Cannabis* cultivars exposed to W20/4 and W24/0. Moreover, this would also explain the decline in RE_0_/RC since in that way pressure on PSI would be reduced. It was suggested that the development of higher dissipation, in the form of heat, usually indicates formation of a protective mechanism against photooxidative damage that could be induced by absorption of unnecessary excitation energy [28]. Similar inactivation of RCs was reported for photoinhibited common fig leaves exposed to high light [60] as well as for ornamental plants grown under continuous light [61].

Purple light induced a different response. When both cultivars were exposed to P20/4, all fluxes increased with the exception of the RE_0_/RC parameter that remained at the same level. However, the exposure to P24/0 in Finola induced the increase in ABS/RS, DI_0_/RC, and RE_0_/RC, while in USO31 it led to a decrease in TR_0_/RC followed by an increase in DI_0_/RC. As already discussed, the increase in all energy fluxes could be the result of formation of Q_A_-non-reducing reaction centers that are unable to efficiently utilize all of the absorbed energy but to dissipate a certain part in form of heat and to use a certain part in electron transport further than Q_A_^−^—what would explain the increased ET_0_/RC. Moreover, CVA analysis confirmed that reactions in USO31 at P20/4 (Figure 3B) were indeed most influenced by the ABS/RC, TR_0_/RC, and DI_0_/RC parameters (Table 1).

The CVA revealed connections between the specific parameters as well as between the parameters and treatments (Figure 3A,B). Selected parameters and specific treatments explained 78.6% and 94.3% of the total variation in Finola and USO31 cultivars, respectively, in first two canonical variates. Other canonical variates explained less than 10% of the total variation, so they were not analyzed further in the manuscript. Reactions of Finola (Figure 3A), positioned on the right side showed that reactions of W16/8 were mostly determined by efficient reactions at the PSI (Table 1), corroborated by the analysis of the coefficient weights of the first two variates (Table 1), where the highest coefficients were observed for ET_0_/ABS and ET/TR_0_ in the first two canonical variates. Similar reactions were observed for the USO31 cultivar (Figure 3B) also with positioning of the W16/8 treatment on the right side of the biplot, however, with highest leverage of RE_0_/RC representing the quantum yield for reduction of end electron acceptors at the PSI acceptor side. On the other hand, energy fluxes per active RC mostly influenced reactions in P20/4 treatments of both cultivars.

The detected grouping showed that duration of exposure as well as the type of light might have influenced the photosynthetic reactions in *Cannabis* cultivars revealing different types of response to prolonged light exposure. To further analyze this matter, based on canonical variate analysis, the Mahalanobis distance between groups was calculated (Table 2). The largest distance in both cultivars was observed between P20/4 and W16/8 light exposures, with *p* < 0.001. The second largest distance in Finola was observed between W16/8 and the other two purple light treatments, while in USO31, the second highest distance was between W16/8 and W20/4, indicating a more sensitive photoperiodic response.

### 2.3. Performance Index (PI_total_) and Partitioning of Total Driving Forces

The performance index for energy conservation from the absorption all the way to the reduction of PSI end acceptors (PI_total_) is well known as the most sensitive parameter since it includes most important functional steps of primary photochemistry, and consequently the vitality of photosynthetic units [38,45,59]. It can be seen that growing under white light and the shortest photoperiod (16/8) induced the highest values in both cultivars (Figure 4A,B, inserts), significantly higher compared to plants grown under purple light. Growth at prolonged photoperiods (20/4 and 24/0) provoked somewhat similar responses, regardless of type of light, with the 20/4 treatment being responsible for the most intense decrease in PI_total_. However, both total and partial driving forces (Figure 4A,B) for photosynthetic activity (DF_total_) revealed that different photoperiods and light types induced diverse responses that led to the decrease in PI_total_. Moreover, greater differences observed were provoked by the longest exposure to light (24/0) rather than type of the light itself. Namely, the decrease in PI_total_ at W20/4 in both cultivars came from the negative difference in all four partial driving forces, although in USO31, those negative differences were greater compared to the control. Such results suggested that all four partial DFs contributed equally to the decrease in PI_total_ at W20/4. However, the contribution of log δ_R0_/(1 − δ_R0_) seems to be greater compared to the other three partial DFs. The P20/4 treatment again showed a similar response in Finola and USO31 cultivars, with a more negative value for log δ_R0_/(1 − δ_R0_) in Finola. This result indicates that at 20/4 exposure, the contribution of reactions involving PSI [45] was lower than in the control and that those reactions were responsible for the PI_total_ decrease.

At continuous light exposure (24/0), white light induced a decrease in PI_total_ for both cultivars, but those values were higher than those at 20/4. However, reactions that caused a decrease were somewhat different in Finola and USO31. In Finola, the decrease in PI_total_ at W24/0 came mainly from negative log ψ_E0_/(1 − ψ_E0_) and positive contribution from log _δR0_/(1 − _δR0_) compared to the control. Negative ψ_E0_/(1 − ψ_E0_) describes the contribution of the dark reactions to the potential for energy conversion between PSII and PSI [38,45], suggesting lower electron transport between two photosystems in W24/0 than in control. However, at P24/0, a decrease in all partial DFs contributed to the decrease in PI_total_, again with the greatest contribution from log δ_R0_/(1 − δ_R0_). USO31 showed at W24/0 a similar contribution of three partial DFs, namely log φ_P0_/(1 − φ_P0_), log ψ_E0_/(1 − ψ_E0_), and log δ_R0_/(1 − δ_R0_), while the log γ_RC_/(1 − γ_RC_) contributed the least. On the other hand, PI_total_ at USO31 P24/0 showed an increase, although not significant, compared to the control. Such increase came from the positive contribution from log δ_R0_/(1 − δ_R0_) confirming the highly efficient PSI electron transport as a physiological adaptation to stressful conditions.

Moreover, the CVA analysis (Figure 3A,B) and the inspection of the coefficient weights (Table 1) confirmed that log γ_RC_/(1 − γ_RC_) and log δ_R0_/(1 − δ_R0_) considerably determined photosynthetic reactions induced by W24/0 treatment in both cultivars. On the contrary, log φ_P0_/(1 − φ_P0_) and log ψ_E0_/(1 − ψ_E0_) influenced the negatively photosynthetic reaction induced by P24/0 treatment in both cultivars. It could be also observed that photosynthetic activity in W16/8 in both cultivars was also mostly determined by log γ_RC_/(1 − γ_RC_) and log δ_R0_/(1 − δ_R0_).

### 2.4. TBARS Content as a Measure of Oxidative Stress

The level of lipid peroxidation was expressed as TBARS content (Figure 5). There is a general trend of TBARS content increase with the duration of light exposure, however, it seems that type of light had significant role. Therefore, exposure to white light at any photoperiod caused an increase in TBARS content. Purple light on the other hand, in the combination with the 20/4 photoperiod revealed even lower values in Finola compared to the control while in USO31 there was no significant change. Prolonged exposure (24/0), however, induced a significant increase compared to controls in both cultivars, regardless of the light type.

The extent of lipid peroxidation is often used as a measure of thylakoid membrane damage under stress conditions [62,63]. Although high light is usually the primary cause of photoinhibition, even weak illumination can cause photoinhibition on the donor side of PSII inducing the formation of reactive oxygen species (ROS) [62,64]. In such conditions, ROS can instantly react with proteins and associated lipids on the donor side of PSII, causing damage and initiating lipid peroxidation [63]. Moreover, it was reviewed that prolonged illumination could induce the same response as very high light intensities and that the generation of ROS, especially of ^1^O_2_, increases exponentially with time [19]. This could explain the increased damage of lipids at continuous exposure (24/0) to light in our investigation, regardless of light type.

Constant high light was shown to cause photoinhibition of photosystems due to the photodamage. However, the repair cycle of PSII was shown to be very refined and it can rapidly restore PSII activity [64]. On the other hand, PSI photoinhibition is not necessarily associated with high light intensity. When PSI photoinhibition occurs, cyclic electron transport is created that serves as a protection of PSI since it redirects electrons into the plastoquinone pool and reduces pressure to PSI [65]. Increased oxygen reduction on the PSI reducing side activates two enzymes, namely catalase and ascorbate peroxidase, to compensate for oxygen reduction. This activates the water–water cycle that serves as a dissipation mechanism of excess reducing power [19,63,65]. In that case, increased scavenging of ROS could alleviate damage to lipids and, hence, decrease products of lipid peroxidation. Since our results revealed a positive G-band in both cultivars at P20/4 (Figure 1E,J), indicating impediments of electron transport at the PSI acceptor side, creation of cyclic electron transport around PSI could be an efficient defense mechanism in conditions of prolonged exposure to purple light. However, continuous light exposure did not provoke such repair mechanism, probably due to the constant exposure to light with no dark periods that would reduce the pressure to PSI.

### 2.5. Evaluation of FT-IR Spectra

Fourier transformation infra-red (FT-IR) spectra analysis revealed additional information on the effect of different photoperiods and light types on changes in molecular structures of macromolecules in *Cannabis* cultivars. It is a versatile analytical method that can provide essential information on the molecular structure of organic compounds in plant extracts [66,67,68]. Biological samples are complex and show broad and diverse absorption peaks in the FT-IR spectra that comes from the vibration of different compounds in the plant sample [68]. Some of the most prominent peaks that can be distinguished in *Cannabis* plants can be seen in the Figure 6A insert. The most pronounced peaks come from vibrations, bending, or stretching of macromolecules when they interact with IR light [67]. The peaks at around 1060 and 1390 cm^−1^ are connected to stretching vibrations which belong to the cell wall polysaccharides while the peak around 1590 cm^−1^ belongs to the lignin. The peaks between 3000 and 2800 cm^−1^ represent lipid stretching vibrations and the peak around 3400 cm^−1^ is showing stretching vibrations that correspond to the proteins [67,68]. Since many bands with specific structures in the sample usually overlap with each other, PC analysis was used to distinguish specific clustering. It was recently reported that cluster analysis could serve as a good tool for separation of species and cultivars, especially when the differentiation depends on various unknown criteria [66]. The PC analysis (Figure 6B) gave two principal components with eigenvalues greater than 1, expressing more than 70% of the total variance. PCA provides the results given in Figure 6A for the scores (i.e., objects, samples). It shows 12 *Cannabis* samples in the 2D figure with one of the two components on each axis. Two main clusters of samples can be distinguished: a four-members cluster on the left side which contains only those samples grown at 16/8 photoperiod regardless of light type and a rather compact cluster on the right side which contains the samples of both cultivars grown under white (W) light. As evident in the PC score plot, the different hemp samples can be divided into two distinct groups, which would suggest that they have different susceptibilities to the type and duration of light exposure.

## 3. Materials and Methods

### 3.1. Plant Material and Experimental Approach

Seeds of two industrial hemp (*Cannabis sativa* subsp. *sativa*) cultivars (Finola and USO31) were presoaked for 48h in tap water and then seeded into the mixture of commercial soil (Klasman TS2) and perlite in a 70:30 ratio, respectively. Plants were cultivated in growth chambers with artificial LED light (~80 μmol m^−2^s^−1^, 22 ± 2 °C) until the appearance of the first real leaves. Week-old seedlings were transferred into the fresh soil (the same as for sowing) in separate plastic pots (φ 25 cm). Five randomly chosen plants of each cultivar were placed into two separate chambers (~250 μmol m^−2^s^−1^, 22 ± 2 °C), one with purple light (LED lighting, VGD Lumia, Mouans-Sartoux, France) and one with white light (QMH and LED lighting, Vötsch V7014, Balingen, Germany). Three separate experiments were conducted with three photoperiods, 16/8, 20/4, and 24/0 h of day/night cycle, where the 16/8 photoperiod was considered as a control. Plants were watered regularly as needed. Five-week-old plants were used for fast chlorophyll *a* fluorescence measurements (ChlF), determination of lipid peroxidation (TBARS), and FT-IR analysis.

### 3.2. Chlorophyll a Fluorescence Measurements

For fast ChlF transients, the second leaves (first fully grown) from the top of each plant were measured (n = 5) using Handy PEA (Hansatech Instruments Ltd., King’s Lynn, UK). Leaves were dark adapted for 30 min before the measurement. The ChlF transients were induced with a pulse of saturating red light for 1 s (3200 μmol m^−2^s^−1^, peak at 650 nm) and then analyzed using the JIP test [59,69,70,71]. The OJIP transients were obtained as mean values of five measurements per cultivar and treatment. Selected structural and functional parameters (Table 3) calculated from the JIP test were used to evaluate the condition of the photosynthetic apparatus in both cultivars grown under two light types and three different photoperiods (16/8, 20/4, and 24/0). Each cultivar grown under white (W) and purple light (P) was normalized to their respective control (16/8 photoperiod). OJIP transients for specific events in the OK, OJ, JI, and IP phases were calculated and presented as difference ΔV_OP_, ΔV_OK_, ΔV_OJ_, ΔV_JI_, and ΔV_IP_ normalized to the controls [38]. Total driving force (DF_total_) of the total photosynthetic electron transport, calculated as log PI_total_, was summed up using the four corresponding partial driving forces: log γ_RC_/(1 − γ_RC_), log φ_P0_/(1 − φ_P0_), log ψ_E0_/(1 − ψ_E0_), and log δ_R0_/(1 − δ_R0_), while the difference for the 20/4 and 24/0 photoperiods for each cultivar. The ∆DF_total_ were calculated as ∆DF_total_ = DF_20/4 or 24/0_ − DF_16/8_ [45,72].

### 3.3. Determination of the Lipid Peroxidation Level

The amount of thiobarbituric acid reactive substances (TBARS) from the TBA reaction was used to determine the level of lipid peroxidation [73]. The absorbance was measured at 532 nm (Specord 40, Analytik Jena, Jena, Germany) and the value for non-specific absorption at 600 nm was subtracted. To calculate the concentration of TBARS, an extinction coefficient of 155 mM^−1^ cm^−1^ was used.

### 3.4. Fourier Transform Infrared Spectroscopy (FT-IR) Measurements

To screen two *Cannabis* cultivars grown under white and purple light and different photoperiods, FT-IR spectroscopy was used. Approximately 10 g of powdered leaves was extracted with 50 mL of ethanol and vigorously stirred on a mechanical shaker for 3 h. The supernatants were collected and dried and then mixed with KBr. Twelve absorbance spectra were measured between 400 and 4000 cm^−1^ spectrophotometrically (FTIR-8400S, Shimadzu, Tokyo, Japan). The selected FT-IR spectrum is shown in Figure 6A, together with the characteristic values of the most pronounced maxima.

### 3.5. Data Analysis

Statistical analyses for cultivars grown at 16/8, 20/4, and 24/0 photoperiods in white and purple light, respectively, were performed using Statistica software (ver. 13.1., Tibco Software Inc., Palo Alto, CA, USA). To analyze ChlF parameters and the level of lipid peroxidation, an analysis of variance (ANOVA) followed by the Least Significant Difference (LSD) *post-hoc* test was performed. Differences were considered significant at *p* < 0.05. The results are presented as mean ± standard deviation (SD) of five replicates (n = 5). The overall variation in the JIP parameters was examined using the canonical variate analysis (CVA). CVA aims to maximize between-group distances within latent space (canonical axes/variates). The basis of this space is the principal components of a PCA computed from the among-groups covariance matrix premultiplied by the inverse pooled within-group covariance matrix [74]. Further, based on the results of CVA and group designations, the Mahalanobis distance was calculated, along with distance *p*-values based on 1000 permutations in the R/Morpho package. Besides visual inspection of the positions of FT-IR absorption maxima, the overall variation in the data was examined using a Principal Component Analysis (PCA). Through reduction of multidimensional variable-space, the PCA allows us to render a visual display of overall variation in the complex dataset. The PCs are extracted so that the first principal component (PC1) accounts for the largest amount of the total variation in the data, PC2 accounts for the maximum amount of the remaining total variation not already accounted for PC1, etc.
PC_i_ = *l*_1i_*X*_1_ + *l*_2i_*X*_2_ + … + *l*_ni_*X*_n_
where PC*_i_* is the *i*th principal component and *l_ni_* is the loading of the observed variable *X_n_*.

## 4. Conclusions

Based on the results obtained from this investigation, it can be concluded that different photoperiods and different types of light provoked adverse responses of two *Cannabis* cultivars. The 16/8 cultivation under white light was proven to create the most beneficial conditions for efficient physiological response corroborated with the highest PI_total_ in both cultivars. Although the 16/8 cultivation under purple light revealed lower PI_total_ values compared to those measured under white light, both types of light showed lowest TBARS contents suggesting the lowest level of thylakoid membrane damage. After exposure to a prolonged photoperiod (20/4) and continuous light (24/0), cultivation under white light (W) induced similar responses in both cultivars. White light, at both photoperiods, caused inactivation of certain parts of RCs causing lower ability for regulation of absorbed energy and its utilization by photosynthetic membranes. However, higher dissipation of excess light (DI_0_/RC) reduced the pressure on PSI which caused reduced capacity for photochemical Q_A_ reduction and consequently caused a slower transfer of electrons to the PSI acceptor side (negative G-band). Therefore, exposure to white light most probably increased the number of available NADPH^+^ molecules which served as an advantageous compensatory mechanism. Purple light, however, provoked somewhat different responses depending on the duration of exposure to light. Although the 20/4 cultivation caused inactivation of OEC and revealed a poor ability for regulation of absorbed energy and its utilization, alternative electron donors maintained the electron feed from the OEC to the RCs. In addition, efficient dissipation of excess energy and formation of cyclic electron transport around PSI suggested that purple light initiated competent photoprotective strategies and promoted the efficient repair system (lower TBARS level). Cultivation under continuous purple light (P24/0) retained the functional electron transport between two the photosystems and increased dissipation ultimately maintained PI_total_ suggesting a positive effect on the photosynthetic reaction even though damage to thylakoid membranes occurred. Additional analyses of antioxidant activity and antioxidant response are needed to elucidate the detailed physiological profiles of two *Cannabis* cultivars due to prolonged light exposure and type of light.

## Figures and Tables

**Figure 1 ijms-23-09702-f001:**
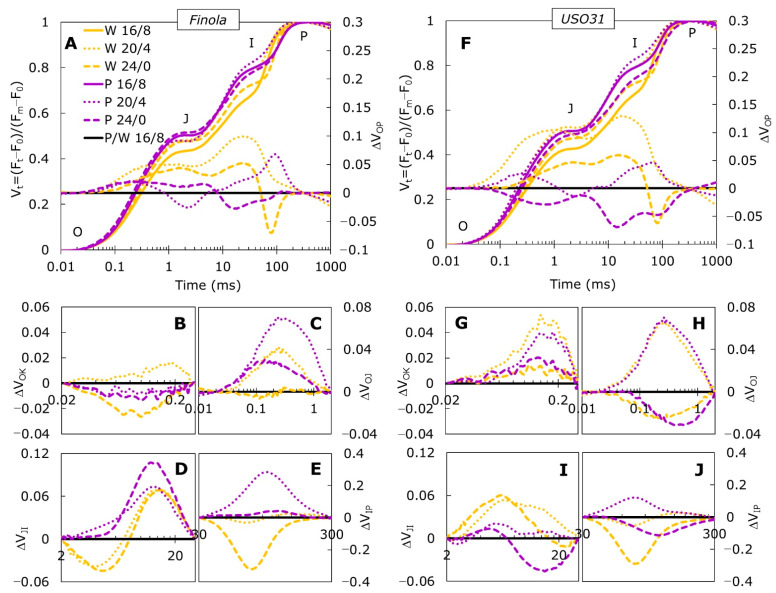
Differences in the shapes and amplitudes of OJIP transient curves measured in *Cannabis* cultivars Finola and USO31 exposed to 16/8, 20/4, and 24/0 photoperiods and grown under white and purple light are presented as kinetics of relative variable fluorescence V_t_ and as difference kinetics ΔV_OP_ (**A**,**F**). Difference kinetics ΔV_t_, for individual bands, L (**B**,**G**), K (**C**,**H**), H (**D**,**I**), and G (**E**,**J**) are plotted at different time ranges. Each curve represents the average of five measurements (n = 5) per treatment. The 16/8 photoperiods were used as referent values for each cultivar and growth light. The O, J, I, and P steps are indicated in Vt curves.

**Figure 2 ijms-23-09702-f002:**
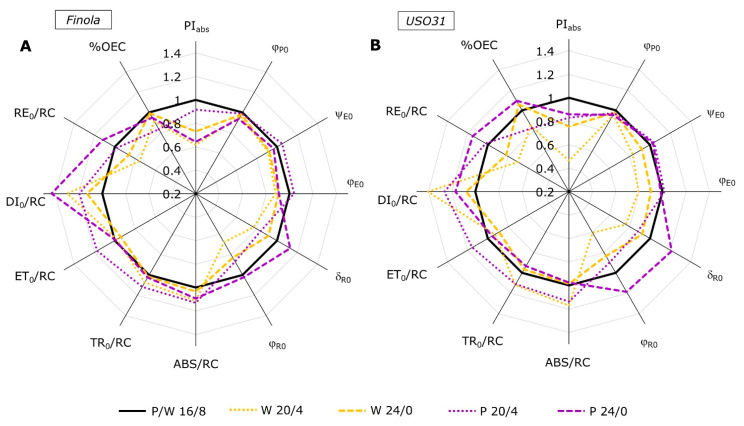
Radar plot of selected JIP-test parameters characterizing PSII functioning measured in *Cannabis* cultivars Finola (**A**) and USO31 (**B**) exposed to 16/8, 20/4, and 24/0 photoperiods and grown under white and purple light. Data are normalized to their respective controls measured at the 16/8 photoperiod for each cultivar and growth light separately (control = 1). Each curve represents mean values of 5 measurements (n = 5); LSD_Finola_ = 0.019; LSD_USO31_ = 0.008.

**Figure 3 ijms-23-09702-f003:**
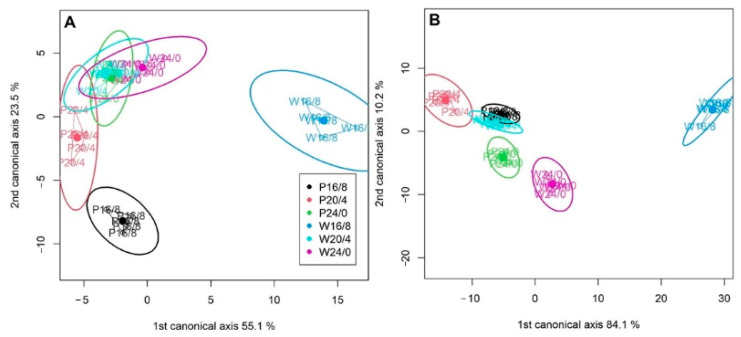
Cannonical variate analysis (CVA) representing the explanatory variables within *Cannabis* cultivars Finola (**A**) and USO31 (**B**) exposed to 16/8, 20/4, and 24/0 photoperiods and grown under white (W) and purple (P) light. Points represent values for each treatment. Ellipses around the different groups represent the 95% confidence intervals.

**Figure 4 ijms-23-09702-f004:**
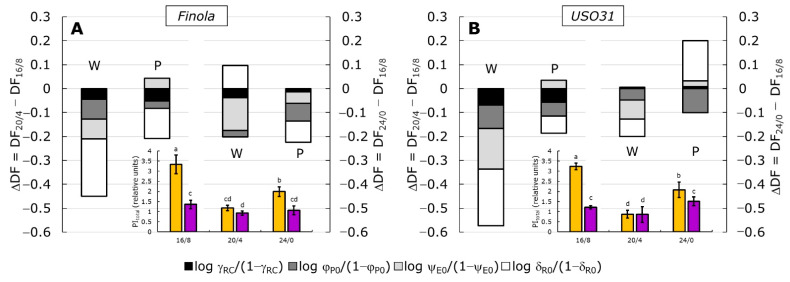
Relative changes in the difference in driving forces (ΔDF) measured in *Cannabis* cultivars Finola (**A**) and USO31 (**B**) exposed to 16/8, 20/4, and 24/0 photoperiods and grown under white (W) and purple (P) light. Stacked columns are showing the difference in DFs at 20/4 (left columns per panel) and 24/0 (right columns per panel) photoperiods minus the DF at 16/8 (control). Each DF is calculated by summing up their partial driving forces: log γ_RC_/(1 − γ_RC_), log φ_P0_/(1 − φ_P0_), log ψ_E0_/(1 − ψ_E0_), and log δ_R0_/(1 − δ_R0_). The performance index for energy conservation from exciton to the reduction of the final electron acceptor at PSI, PI_total_ for each cultivar is shown in the inserts. Yellow bars represent plants grown under white light, while purple bars represent plants grown under purple light. Results are shown as the mean of five independent measurements (n = 5) ± SD. Different letters represent a significant difference at *p* ≤ 0.05 (ANOVA, LSD).

**Figure 5 ijms-23-09702-f005:**
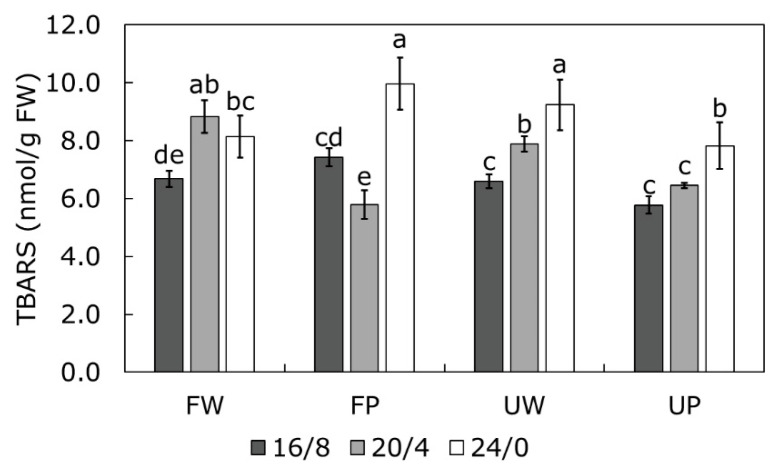
Changes in content of thiobarbituric acid reactive substances (TBARS, nmol/g FW) measured in *Cannabis* cultivars Finola (F) and USO31 (U) exposed to 16/8, 20/4, and 24/0 photoperiods and grown under white (W) and purple (P) light. Results are shown as mean of five independent measurements (n = 5) ± SD. Different letters represent significant differences at *p* ≤ 0.05 (ANOVA, LSD); LSD = 0.187.

**Figure 6 ijms-23-09702-f006:**
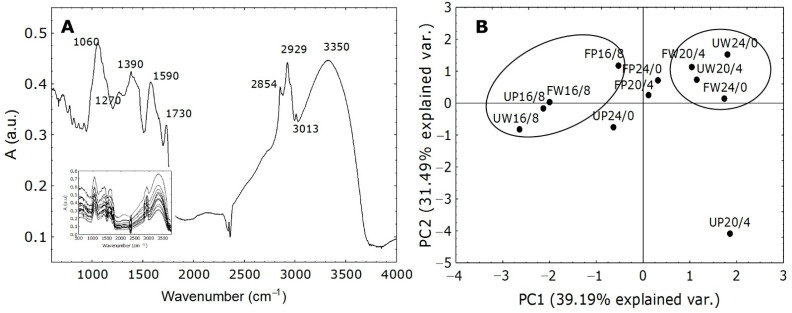
Characteristic bands (range 400–4000 cm^−1^) of FT-IR spectra measured in the *Cannabis* samples. (**A**) shows characteristic peaks marked with numbers. The FTIR spectrum was recorded using 12 scans at a resolution of 2 cm^−1^ in the wavenumbers range from 400 to 4000 cm^−1^. The insert represents the FTIR spectral analysis in *Cannabis* cultivars for each treatment. PCA (**B**) represents the explanatory FT-IR variables between and within *Cannabis* cultivars Finola (F) and USO31 (U) exposed to 16/8, 20/4, and 24/0 photoperiods and grown under white (W) and purple (P) light. Points represent values for each treatment. The analysis provides the score results provided in the insert.

**Table 1 ijms-23-09702-t001:** Coefficients of the first two canonical variates (Figure 3) in Finola and USO31 cultivars.

Parameter	Finola		USO31	
	CV1	CV2	CV1	CV2
ABS/RC	−54.3	154.9	−212.3	−310.6
DI_0_/RC	−407.8	307.9	−228.3	−465.6
TR_0_/RC	353.6	−153.0	16.0	155.1
ET_0_/RC	−156.2	−95.2	681.3	685.8
RE_0_/RC	232.1	189.6	−87.5	189.7
φ_P0_	−1016.8	−936.9	−1565.2	3258.2
ψ_E0_	2373.0	−5883.1	−745.5	8882.0
φ_E0_	−3768.5	7400.1	−2460.7	−13,393.0
δ_R0_	−408.7	−256.3	−1348.0	947.1
φ_R0_	260.5	−327.2	4068.7	−2938.0
γ_RC_/(1 − γ_RC_)	682.7	1803.5	−870.0	−1354.0
φ_P0_/(1 − φ_P0_)	−24.8	−0.1	−76.0	−78.6
ψ_E0_/(1 − ψ_E0_)	−7.0	273.9	−375.8	−282.4
PI_abs_	153.3	−197.7	323.2	314.4
δ_R0_/(1 − δ_R0_)	14.3	35.6	48.0	−109.7
PI_total_	−9.1	6.2	−75.1	79.2

**Table 2 ijms-23-09702-t002:** Mahalanobis distances in Finola and USO31 cultivars exposed to 16/8, 20/4, and 24/0 photoperiods and grown under white (W) and purple (P) light. * represents significance at *p* < 0.05, ** represents significance at *p* < 0.01, *** represents significance at *p* < 0.001, based on 1000 permutations.

Finola					
	P16/8	P20/4	P24/0	W16/8	W20/4
P20/4	11.18 *				
P24/0	11.96 **	8.36			
W16/8	18.32 ***	19.88 ***	17.31 ***		
W20/4	12.47 **	8.32	3.49	17.8 ***	
W24/0	12.98 **	12.66 **	8.34	16.77 ***	9.12
**USO31**					
P20/4	11.1				
P24/0	9.4	13.64			
W16/8	33.91 ***	42.47 ***	34.07 ***		
W20/4	8.64	11.86	8.04	34.83 ***	
W24/0	14.44	21.75 *	10.13	28.07 **	15.04

**Table 3 ijms-23-09702-t003:** Calculations and definitions of selected JIP-test parameters [38,59,69,70,71].

Recorded Data and Technical Parameters	Description
F_0_	Minimal fluorescence intensity (20 μs)
F_m_	Maximal fluorescence intensity
V_J_ = (F_J_ − F_0_)/(F_m_ − F_0_)	Relative variable fluorescence at 2 ms
V_I_ = (F_I_ − F_0_)/(F_m_ − F_0_)	Relative variable fluorescence at 30 ms
V_K_ = (F_K_ – F_0_)/(F_m_ – F_0_)	Relative variable fluorescence at 0.3 ms
F_V_ = F_m_ − F_0_	Maximal variable fluorescence
M_0_ = (d*V*/d*_t_*)_0_	Approximated initial slope of relative variable fluorescence F_v_
φ_P0_ = TR_0_/ABS = F_v_/F_m_	Maximum quantum yield of PSII
ψ_E0_ = ET_0_/TR_0_ = 1 − V_J_	Probability that a trapped exciton moves an electron further than Q_A_
φ_E0_ = ET_0_/ABS = [1 − (F_0_/F_m_)](1 − V_J_)	Quantum yield for electron transport
δ_R0_ = RE_0_ − ET_0_ = (1 − V_I)_/(1 − V_J_)	Probability with which an electron from the intersystem electron carriers moves to reduce end electron acceptors at the PSI acceptor side
φ_R0_ = RE_0_/ABS = [1 − (F_0_/F_m_)]ψ_E0_ δ_R0_	Quantum yield for reduction of end electron acceptors at the PSI acceptor side
%OEC = [1 − (V_K_/V_J_)]_treatment_/[1 − (V_K_/V_J_)_control_	Fraction of Oxygen Evolving Complexes (OEC)
ABS/RC = M_0_(1/V_J_)[1/φ_P0_]	Absorption flux per active RC
TR_0_/RC = M_0_(1/V_J_)	Trapping flux per active RC
ET_0_/RC = M_0_(1/V_J_)(1 − V_J_)	Electron transport flux per active RC
DI_0_/RC = (ABS/RC) − (TR_0_/RC)	Dissipation flux per active RC
γRC = Chl_RC_/Chl_total_ = RC/(ABS + RC)	Electron flux reducing end electron acceptors at the PSI acceptor side, per RC
RE_0_/RC = M_0_(1/V_J_)ψ_E0_ δ_R0_	Probability that a PSII Chl molecule functions as RC
PI_ABS_ = [γ_RC_/(1 − γ_RC_)][φ_P0_/(1 − φ_P0_)][ψ_E0_/(1 − ψ_E0_)]	Performance index on an absorption basis
PI_total_ = PI_ABS_[δ_R0_/(1 − δ_R0_)]	Performance index for energy conservation from exciton to the reduction of the PSI end acceptor

## Data Availability

All datasets for this study are included in the manuscript file.

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
