# Peer review of "Effect of Prolonged Photoperiod on Light-Dependent Photosynthetic Reactions in Cannabis"

_ijms, 2022, doi:10.3390/ijms23179702_

Round 1

Reviewer 1 Report

The draft by Gajdosik et al. (2022) titled “Effect of prolonged photoperiod on light-dependent photosynthetic reactions in Cannabis” was assessed. The authors have carried out an interesting work on the photoperiod in Cannabis.Summary, this section should end: “These findings indicate that........”. Good Introduction (current, clear and well written). Line 47 cannabidinol (CBD). Lines 92 and 94, singlet oxygen(1O2). This draft contains a good number of very current references related to the subject under study. Which I value very positively. The references must be checked conveniently. That is, the authors must follow the instructions of Int. J. Mol. Sci. (IF5.542) (e.g. journals must be written in abbreviation). Regarding Results andDiscussion, the authors have constructed a very well written and scientifically correct section. It is interesting to highlight the quality of the figures in terms of their degree of information. Let us think that the topic of this work has a certain complexity. That is, the determination of the parameters involved in the light phase of photosynthesisis highly complicated. Line 434,....measured in in Cannabis... Line 471,....Cannabis cultivars for.... Mat & Met., line 479 ...experimental approach. The rest, clear and well written. Interesting Tab. 1. Conclusions: explain the two sentences corresponding to lines 560-564.

Author Response

Responses to Reviewer 1

Thank you very much for such positive feedback about our work. We appreciate your notion of the complexity of the subject and the quality of the presented result. Also, thank you for the suggestions that will further improve our work. For the list of all changes we made, please see the attachment.

Sincerely,

Selma Mlinarić

Reviewer 2 Report

Manuscript "Effect of prolonged photoperiod on light-dependent photosynthetic reactions in Cannabis" is very interesting.

General comments:

Authors determined the effect of different durations of light exposure as well as different types of light on primary photosynthetic reaction complemented with the level of lipid peroxidation as a measure of oxidative stress in two Cannabis cultivars.

Detailed comments:

Figure 1 needs regression models and coefficients of determination.
Figure 2 needs LSD values.
Figure 5 needs LSD values.
Figure 6B needs PCs values.

My suggestion:

Principal Component Analysis is incorrect for data with replications. Authors should use the Canonical Variate Analysis and estimate the Mahalanobis distances.

Paper needs major revision.

Author Response

Responses to Reviewer 2

Thank you for positive feedback about our work. For the all proposed changes we made, please see the attachment.

Sincerely,

Selma Mlinarić

Round 2

Reviewer 2 Report

PCA is based on average values. By using this methods, we lost the information contained in the replications. The CVA provides more complete information. The Mahalanobis distances give information about differentiation from all the traits taken together.

Paper needs major revision.

Author Response

Dear reviewer,

According to your valued suggestion, we carried out the Canonical variate analysis and computed Mahalobis distances, along with their p values in 1000 permutation test i R/Morpho package. Accordingly, new Figure 3 was added (now on Page 9), Table 1 was corrected (now on Page 10) based on newly performed analysis and newly derived Table 2 was added (page 11). Figure and Table captions were corrected as well as the following text and analysis description in M&M section (Page 17). The reference list was also complemented accordingly.

Although the results appear similar as those obtained by the PCA analysis, the differences between treatments are now more pronounced, and precisely quantified. Thank you for your valuable and constructive contribution.

Best regards,

Selma Mlinarić

Round 3

Reviewer 2 Report

Figure 3A: first canonical variate ranged from ~-7 to ~17. However, (for this same analysis!!!???) in Table 1: from -408.7 to 3768.5.
Similarly for Figure 3B and Table 1.
Authors should get in touch with statistician.
Paper needs major revision.

Author Response

Briefly, the Reviewer 2 gave several general comments in the first round of review, along with one regarding principal component analysis (PCA) that was addressed by a statistician in the first revised manuscript version (and was added as new coauthor).
The Reviewer further considered the whole methodology of PCA unfit for analysis of replicated JIP-test data, in dispute to the whole community using the methodology for multivariate analysis and the referenced statistical handbooks in the first round of the revisions. Reviewer for some reason considered canonical variate analysis (CVA) with Mahalanobis distance would be more appropriate. After the CVA was carried out as suggested, along with Mahalanobis distance and reported in the second revised manuscript version in Figure 3A
and B, and in Table 1 and 2, Reviewer 2 is apparently unable to distinguish coefficients from the canonical variates in the report, putting his authority to suggest statistical methods to question, along with his benevolence in this review process.
We thus seek your assistance to finalize the review process, as all the
reviewer's suggestions were accepted, and all changes were carried out with point-to-point responses.